# Two Cases of Rare Manifestations Due to *Neisseria meningitidis* During the Post-COVID-19 Era in Greece

**DOI:** 10.3390/microorganisms13092071

**Published:** 2025-09-05

**Authors:** Kalliopi Avgoulea, Genovefa Chronopoulou, Athanasia Xirogianni, Stelmos Simantirakis, Theano Georgakopoulou, Anastasios Tsakalos, Constantinos Karamalis, Lampros Nikolopoulos, Fotios Roussos, Maria Gryllia, Nektarios Marmaras, Efterpi Oikonomou, Diagoras Zarganis, Maria Orfanidou, Anastasia Pangalis, Muhamed-Kheir Taha, Georgina Tzanakaki

**Affiliations:** 1National Meningitis Reference Laboratory, Department of Public Health Policy, School of Public Health, University of West Attica, 196, Alexandras Avenue, 11521 Athens, Greece; k.avgoulea@gmail.com (K.A.); axirogianni@uniwa.gr (A.X.); ssimantirakis@uniwa.gr (S.S.); kkaramalis@uniwa.gr (C.K.); 2Laboratory of Clinical Microbiology, General Hospital of Athens “G. Gennimatas”, 154, Mesogeion Avenue, 11527 Athens, Greece; tastsak82@gmail.com (A.T.); mariacorf@gmail.com (M.O.); 3Central Laboratory, Athens Medical Center, Distomou 5-7, 15125 Marousi, Greece; g.chronopoulou@iatriko.gr (G.C.); n.marmaras@iatriko.gr (N.M.); a.pangalis@iatriko.gr (A.P.); 4Department of Vaccine Preventable Diseases, National Public Health Organization (NPHO), 15123 Athens, Greece; t.georgakopoulou@eody.gov.gr; 5Neurology Department, General Hospital of Athens “G. Gennimatas”, 154, Mesogeion Avenue, 11527 Athens, Greece; lamprosnikolo@gmail.com (L.N.); roussosfotios@gmail.com (F.R.); gryllia@otenet.gr (M.G.); 6Second Pediatric Department, Athens Medical Center, Distomou 5-7, 15125 Marousi, Greece; oikonomouefterpieleftheria@gmail.com (E.O.); diagoraszarganis@gmail.com (D.Z.); 7Invasive Bacterial Infections, Institut Pasteur, 25-28 Rue du Docteur Roux, 75015 Paris, France; muhamed-kheir.taha@pasteur.fr

**Keywords:** *N. meningitidis*, atypical manifestation, MenX, MenB, MLST, ciprofloxacin resistance

## Abstract

**Background/Objectives**: *Neisseria meningitidis* is a human-specific pathogen capable of causing life-threatening illnesses. Occasionally, it is recovered from unusual sites, other than the bloodstream or the central nervous system. Herein, we describe two rare manifestations due to *N. meningitidis* within a year (2024) in Greece. **Methods**: Atypical infection due to *N. meningitidis* was diagnosed in two different patients: *Case-1* presented with an inflammatory swelling in the mid-line of the neck, and *Case-2* presented with swelling of the left knee. Both patients had high fever and no neurological signs at admission; *Case-2* progressed to meningoencephalitis. Phenotypic and genotypic identifications were carried out in both cases. **Results**: *Case-1* and *Case-2* isolates were identified as follows: MenX: 18, 25-44, F5-5, ST-823; 198cc and MenB: 7-1, 1, F3-3, ST-7460; 32cc for *Por*A, *Fet*A and MLST, respectively. MenX was identified for the first time in Greece and finetyping revealed rare genotypic characteristics. Both isolates were susceptible to cefotaxime, ceftriaxone and rifampicin, while *Case-2* isolate expressed reduced susceptibility to penicillin and resistance to ciprofloxacin. Both patients recovered fully. **Conclusions**: Although uncommon, *N. meningitidis* may be isolated from atypical sites and specimens. Clinicians and microbiologists should remain aware that meningococcus is a potential cause of infections beyond meningitis and septicaemia.

## 1. Introduction

*Neisseria meningitidis*, also known as meningococcus, is an encapsulated, aerobic, Gram-negative diplococcus belonging to the *Neisseriaceae* family. It is estimated that nearly 10% of the global population carries the bacterium with no symptoms, with carriage rates reaching up to 23.7% among adolescents and young adults in developed countries [1]. Meningococcus is an obligate human pathogen, responsible for the invasive meningococcal disease (IMD) with clinical manifestations such as meningitis and/or septicaemia, which may result in high morbidity and mortality rates worldwide [2].

The pathogen is transmitted via close or lengthy contact with patients or asymptomatic carriers, through dispersion of higher respiratory and throat secretions. Diabetes mellitus, obesity, malignancies, renal disease, immunosuppression, and chronic steroid therapy are all considered as predisposing factors for IMD. However, serious disease manifestations due to *N. meningitidis* have also been described in the immunocompetent population or as a result of the transition from asymptomatic carriage to invasive disease [3,4].

The virulence of *N. meningitidis* is influenced by various factors, such as endotoxin, outer membrane adhesive proteins, iron sequestration, and capsular polysaccharide. Based on the structure and the immune response of the capsule, meningococcus may be classified in 12 serogroups: A, B, C, E, H, I, K, L, W, X, Y, Z, and 29E. The more frequently encountered serogroups responsible for IMD are A, B, C, W, X, and Y. MenB and MenC have the wider geographical distribution, whereas MenX is limited to the African sub-Saharan region. In the industrialized countries, MenB appears to be the predominant cause of endemic meningococcal disease, while MenC seems responsible for both endemic cases and sporadic epidemics [2].

IMD is manifested as acute bacterial meningitis and/or septicaemia, with symptoms and signs such as headache, fever, cervical stiffness, photophobia, severe vomiting, haemorrhagic rash, lethargy, and decreased level of consciousness [5]. At the same time, there is an increasing trend of atypical manifestations due to *N. meningitidis*. Reports on unusual sites of infection include the lower respiratory system [5], soft tissues [6], urogenital tract [7], gastrointestinal tract [5], eyes [8], as well as joints [5].

The aforementioned cases have been described in both adults and children, whereas the transmission route is not always clear. Two rare cases of atypical clinical manifestations caused by *Neisseria meningitidis* are presented. They occurred in Greece, in 2024, in two patients that lacked signs of neurological attack at the time of the diagnosis.

## 2. Materials and Methods

### 2.1. Patients

#### 2.1.1. Case-1

An 8-year-old boy presented at the Emergency Department of a pediatric hospital in Athens in May 2024 with fever lasting 24 h, cough, rhinitis, and a single episode of vomiting. Additionally, purulent discharge was observed in the midline cervical region. According to the patient’s medical history, he had undergone surgical removal of a thyroglossal duct cyst 18 months prior to admission. Over the past three months, the boy had experienced recurrent episodes of inflammation at the surgical site, characterized by localized abscess formation, treated with oral antibiotic therapy, which first consisted of amoxicillin/clavulanic acid, followed by cefprozil.

On clinical examination upon admission, the patient was in relatively good general condition, with a temperature of 37.8 °C and a heart rate of 130 bpm. The clinical presentation included erythematous tonsils and tenderness in the cervical region, while the cervical lymph nodes were palpable and small bilaterally. Clinical examination revealed a localized inflammation in the neck area, including erythema, swelling, and purulent discharge. The neurological assessment was negative, and the purulent fluid was sent to the laboratory for further microbiological investigation.

#### 2.1.2. Case-2

A 25-year-old male visited the Emergency Department of a tertiary general hospital of Athens in November 2024, presenting symptoms such as high fever, swelling of the left knee, and localized pain for the last nine days. The patient’s medical history was not pertinent to his medical condition, and he had no underlying diseases; furthermore, he reported to be vaccinated with MenC vaccine (Meningitec^®^ Pfizer Inc, New York, NY, USA). With the exception of the local inflammatory signs, no other symptoms or signs were recorded, including neurological ones. According to the medical record, the patient was admitted in the Orthopaedics department, where he underwent arthrocentesis. The synovial fluid and blood cultures were sent to the Microbiology Department for further microbiological examination.

Within four hours, the patient’s clinical condition deteriorated acutely, manifesting with symptoms including sudden headache, vomiting, and pain also to the right knee. He was admitted in the Neurology Department, and an emergency lumbar puncture was performed. The cerebrospinal fluid (CSF) was sent to the Microbiology Department for further microbiological tests.

### 2.2. Laboratory Identification

Specimens from both the 8-year-old boy and the 25-year-old male were subjected to Gram-staining for initial assessment, followed by culture on selective and differential media to facilitate the identification of both aerobic and anaerobic bacterial pathogens. For the isolation of fastidious organisms, particularly microaerophilic bacteria, the specimens were inoculated onto chocolate agar—a nutrient-rich medium supplemented with factors V (NAD) and X (hemin)—and incubated at 37 °C in an atmosphere containing 5% CO_2_. In addition, specimens were cultured on Columbia CNA agar (CNA) and CDC anaerobe blood agar to support the growth of Gram-positive cocci and obligate anaerobes, respectively. Plates were incubated under appropriate atmospheric conditions: aerobic, anaerobic, and microaerophilic, depending on the suspected organisms. Identification of isolates was performed using standard microbiological techniques including colony morphology, biochemical testing, and, where appropriate, matrix-assisted laser desorption/ionization time-of-flight mass spectrometry (MALDI-TOF MS), following the manufacturer’s instructions and established protocols.

Additionally, for *Case-2*, the BIOFIRE^®^ FilmArray multiplex PCR system (bioMérieux, Craponne, France) was deployed for timely identification, using Blood Culture Identification 2 (BCID2) and Meningitis/Encephalitis (ME) Panels.

### 2.3. Phenotypic and Genotypic Characterization

Phenotypic and genotypic characterizations of the isolates were conducted at the National Meningitis Reference Laboratory. Serogroup determination was initially performed using a slide agglutination test (Remel Europe Ltd., Dartford, Kent, UK) in accordance with the manufacturer’s instructions. To confirm and further characterize the isolates, a multiplex polymerase chain reaction (PCR) assay targeting capsule-specific genes corresponding to *Neisseria meningitidis* serogroups A, B, C, W, and Y was employed, following previously published protocols [9].

### 2.4. Molecular Characterization and Finetyping

Further molecular characterization of the isolates was performed by finetyping using multilocus sequence typing (MLST), PorA, and FetA gene sequencing, as previously described [10,11]. Sequence analysis was conducted using the Neisseria database hosted at PubMLST.org (http://pubmlst.org/neisseria/) (assessed 20 May 2024 and 14 November 2024) for the two isolates respectively [12]. Sequence types (STs) were assigned based on allelic profiles and grouped into clonal complexes (ccs) according to the database definitions. PorA genotyping focused on variable regions 1 and 2 (VR1 and VR2) and was compared with reference sequences available in the PorA section of the PubMLST database (http://pubmlst.org/neisseria/PorA/) (assessed 20 May 2024 and 14 November 2024) for the two isolates respectively [12]. Similarly, the variable region of the fetA gene (FetA VR) was determined following established protocols [11] and compared with sequences in the FetA VR database (http://pubmlst.org/neisseria/FetA/), accessed on 20 May 2024 and 14 November 2024 for the two patient isolates, respectively [12].

### 2.5. Antimicrobial Susceptibility Testing

Antimicrobial susceptibility testing was performed using E-test gradient strips (LI-OFILCHEM S.R.L., Roseto degli Abruzzi, Italy) in accordance with the guidelines established by the European Committee on Antimicrobial Susceptibility Testing (EUCAST). Minimum Inhibitory Concentrations (MICs) were determined for five antimicrobial agents: penicillin, cefotaxime, ceftriaxone, rifampicin, and ciprofloxacin. Interpretation of MIC values was carried out using the EUCAST Clinical Breakpoint Tables, version 14.0, effective as of 1 January 2024 [13].

## 3. Results

### 3.1. Clinical and Laboratory Data

Both cases presented with fever and a localized swelling, each one in different body sites, with no neurological signs. Laboratory testing for *Case-1* revealed leukocytosis (WBC 18,600/μL) and elevated CRP (58.3 mg/L) (Table 1). Imaging studies, including ultrasound and MRI of the neck, revealed a pathological finding: a linear contrast-enhancing lesion suggestive of a fistulous tract at the level of the hyoid bone, immediately left of the midline, with associated inflammation of the overlying subcutaneous tissue (Figure 1). The patient was diagnosed with a cervical abscess based on clinical and laboratory findings.

*Case-2* laboratory testing upon admission revealed leukocytosis (WBC 21,760/μL) and elevated CRP (295.9 mg/L) (Table 1). Both synovial and cerebrospinal fluids were pathological; the microscopic evaluation by Gram staining revealed the presence of numerous polymorphonuclear neutrophils, along with intracellular and extracellular Gram-negative coffee-bean-like diplococci (Table 2). The blood cultures were negative after five days of incubation.

According to the basic immunological studies (C-3, C-4), both patients were immunocompetent.

### 3.2. Strain Identification by Conventional Methods

Upon 24 h incubation, small, round, greyish, non-haemolytic, glistening colonies with clearly defined edge were observed in both blood and chocolate agar. The isolates were oxidase-positive and displayed the characteristic pink, coffee-bean shape on Gram stain. Ultimately, both isolates were identified as *Neisseria meningitidis*. In Case 2, meningococcal genetic material was additionally detected directly from clinical specimens using the BioFire^®^ FilmArray^®^ Blood Culture Identification Panel 2 (BCID2) and the Meningitis/Encephalitis (ME) Panel (bioMérieux SA, Craponne, France).

### 3.3. Finetyping

Finetyping for *Por*A, *Fet*A and MLST, revealed the following characteristics: MenX: 18, 25-44, F5-5, 198cc (ST-823) and MenB: 7-1, 1, F3-3, 32cc (ST-7460) for *Case-1* and *Case-2*, respectively.

### 3.4. Antimicrobial Susceptibility Testing

Both isolates were susceptible to cefotaxime, ceftriaxone, and rifampicin. Reduced susceptibility to penicillin was found in the *Case-2* isolate (MIC 0.19 mg/L), while it was also resistant to ciprofloxacin (MIC 0.03 mg/L), as opposed to *Case-1* isolate, which was susceptible to both aforementioned antibiotics.

### 3.5. Antimicrobial Treatment and Outcome

*Case-1* treatment included intravenous hydration and empirical antibiotic therapies by the administration of clindamycin and cefotaxime. Patient’s clinical improvement was observed before discharge, with an advice of follow up.

*Case-2* was treated with ceftriaxone 2grx2 iv for 12 days and dexamethasone 8mgx4 iv for 5 days. The patient made a full recovery with no further neurological symptoms and signs.

To our knowledge, chemoprophylaxis was prescribed to the close contacts for both cases.

## 4. Discussion

Atypical presentations, although rare, are increasingly documented, showcasing the changing pattern of infections due to *N. meningitidis*. Pyelonephritis [14], conjunctivitis [15], pericarditis [16], and preseptal cellulitis [17] have been reported more recently. In France, epiglottitis [18], pneumonia, and abdominal forms associated with novel serogroups [5] increased from 2022 onwards and the overall burden of the COVID-19 pandemic is yet to be established. Specifically, to our knowledge, *Case-1*, is the third case reported worldwide as an unusual soft tissue manifestation due to *N. meningitidis*, related to previous surgical removal of a thyroglossal duct cyst. Similar clinical presentations have been documented (2008) in a pediatric patient in the USA [19] and, more recently, (2024) in a 16-year-old boy, otherwise healthy, with thyroglossal duct cyst infection in Thailand [6]. In contrast to the previous two cases, a full strain characterization is presented in *Case-1* [MenX: 18, 25-44, F5-5, 198cc (ST-823)].

When compared to other serogroups, MenX is sporadically associated with invasive meningococcal disease worldwide, while it has also been detected in healthy, asymptomatic carriers (children, adolescents, and young adults) [20,21,22,23,24]. Although MenX has been previously identified in Greece [20], this is the first MenX isolate associated with an atypical presentation in our setting, susceptible to all antibiotics tested.

Although pharyngeal carriage of *N. meningitidis* was not assessed in the child, it is likely that the *Case-1* patient was colonized with meningococcus and subsequently developed a localized infection.

Meningococcal arthritis may be manifested as a monoarticular or multiarticular infection, and the affected joints are various, namely knee [25], hip [26], ankle [27], and elbow [28]. In our study, the disease manifested as monoarthritis, and the affected joint was the knee. Knee joint infections have been reported in both children and adults, with or without positive blood cultures [25,29]. Interestingly, primary meningococcal arthritis has also been reported in the knee joint of a 30-year-old male, along with the presence of calcium oxalate crystals in the synovial fluid [30].

*Case-2* presented as primary septic monoarthritis in an adult male with no neurological signs. This presentation subsequently revealed an underlying IMD with meningitis and polyarthritis. Septic arthritis is a rare but increasingly reported manifestation of *N. meningitidis* worldwide. Primary joint infection is rare, as arthritis usually occurs as a complication of acute meningococcal disease, secondary to invasive meningococcemia, and results directly from meningococcal bacteremia or meningitis [31,32]. In any case, bacteremia is key for the invasive infections by *N. meningitidis* [33]. However, in our study, the blood cultures collected at admission were negative, a finding that might be attributed to an inadequate volume of blood inoculated in the bottles.

Although unusual manifestations of *N. meningitidis* are rare, the most prevalent serogroups causing such presentations are MenW, which is consistently reported with atypical presentations [34], followed by MenY and MenC, and, to a lesser extent, MenB [5,35]. MenY has been indicted as responsible for arthritis in the USA [36] and Italy [37], and MenC has been identified in Romania [38].

The isolate described in this study was typed as MenB, which appears to be responsible for the majority of IMD cases [39]. This is consistent with a recent study from Italy reporting a case of primary knee septic arthritis in an 87-year-old patient, similar to *Case-2* herein presented [29]. Although both the Italian and Greek cases showed similar clinical presentations, the Italian strain belonged to ST-162, cc162, whereas the Greek isolate belonged to ST-7460, cc32.

Susceptibility testing revealed that the isolate exhibited reduced susceptibility to penicillin, a finding not surprising in our setting, given that only 52% of IMD MenB isolates in Greece are susceptible to penicillin [20]. Additionally, the isolate was resistant to ciprofloxacin, an antibiotic recommended for chemoprophylaxis. This case represents the second report of ciprofloxacin-resistant *Neisseria meningitidis* in Greece. The first documented instance was reported in 2020 and involved two MenB isolates belonging to sequence type ST-3129, identified in a migrant camp on the Greek island of Lesbos [40]. An increasing trend of ciprofloxacin resistance has been recently observed in Europe, as well as in East and South Asia, highlighting the importance of continuous surveillance [41].

## 5. Conclusions

In conclusion, *Neisseria meningitidis*, although relatively uncommon in such presentations, can be isolated from atypical sites and clinical specimens, potentially serving as an early indicator of invasive or severe disease. For the prompt initiation of targeted therapy, it is of utmost importance to raise awareness and alertness among both clinicians and microbiologists, ensuring that *N. meningitidis* is considered in the differential diagnosis, even though it is classically associated with meningitis and septicaemia. Moreover, ongoing and continuous surveillance of meningococcal infections is vital for monitoring the evolving epidemiology and for developing tailored strategies and prevention efforts.

## Figures and Tables

**Figure 1 microorganisms-13-02071-f001:**
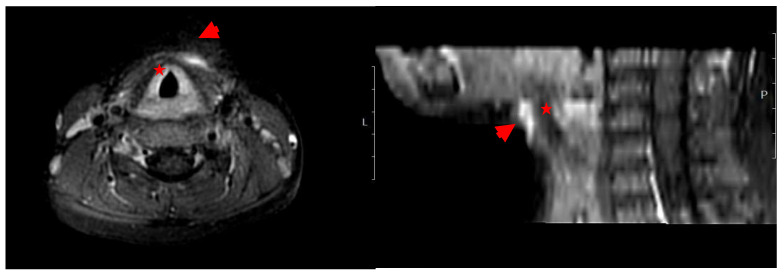
Imaging (MRI). *Case-1*: Characteristics of the lesion in the cervical region. By the use of T2-weighted Magnetic Resonance. Axial (horizontal) view (**left**) and sagittal (side) view (**right**) of the neck. Hyperintense (bright) areas anterior to the trachea (red star) are indicated by red arrowheads suggesting a fluid collection or, abscess, or edematous tissue.

**Table 1 microorganisms-13-02071-t001:** Clinical profile of the patients at admission and overall length of hospitalization.

	Case-1	Case-2
Red Blood Cells (/μL)	4,630,000	5,490,000
**White Blood Cells (/μL)**	**18,600**	**21,760**
**Neutrophils (%)**	**87.4**	**92.30**
Lymphocytes (%)	8.4	3.30
Monocytes (%)	3.9	3.60
Eosinophils (%)	0.1	0.60
Baseophils (%)	0.3	0.20
Platelets (/μL)	416,000	152,000
**C- Reactive Protein (mg/L)**	**58.3**	**295.9**
**Hospitalization days**	**10**	**12**

**Table 2 microorganisms-13-02071-t002:** Microbiological characteristics of the specimens examined in both patients.

	Case-1	Case-2	
**Sample**	**Purulent discharge**	**Synovial fluid**	**Cerebrospinal fluid**
**Gram-stain**	Gram (-) diplococci	Gram (-) diplococci	Gram (-) diplococci
**Oxidase test**	Positive	Positive	Positive
**Culture identification**	*N. meningitidis*	*N. meningitidis*	*N. meningitidis*
**White blood cells (/μL)**	-	40,000	17,000
**Glycose (mg/dL)**	-	0	0.3
**Total protein (mg/dL)**	-	3300	607

## Data Availability

The original contributions presented in this study are included in the article. Further inquiries can be directed to the corresponding author.

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
