# Peer review of "Two Cases of Rare Manifestations Due to Neisseria meningitidis During the Post-COVID-19 Era in Greece"

_microorganisms, 2025, doi:10.3390/microorganisms13092071_

Round 1

Reviewer 1 Report

Comments and Suggestions for Authors

General comments:

This article nicely reports two cases of meningococcal disease with truly unusual presentations, and it should published.

Title:

  1. meningitidis in italic.

Abstract: The authors mentioned in the abstract that both patients had no neurologic symptoms, however, in their results case-2 manifested symptoms and signs of elevated intracranial pressure with a CSF clearly showing a meningoencephalitis, they need to correct that in the abstract. I suggest the authors mention in the abstract the outcomes of both patients, not just the microorganism characteristics.

Materials and Methods: I suggest the authors to explain more about the clinical outcomes such as hospitalization days, CBC findings, etc… They could provide a table for that. In addition, do these patients have any basic immunological studies such as immunoglobulins and complement?

Discussion: There have been many publications about atypical presentations of meningococcal disease, such as conjunctivitis, sinusitis, among many others, I think the authors should point out those publications in their discussion.

Author Response

Title:

meningitidis in italic.

Authors’ reply: DONE

Abstract: The authors mentioned in the abstract that both patients had no neurologic symptoms, however, in their results case-2 manifested symptoms and signs of elevated intracranial pressure with a CSF clearly showing a meningoencephalitis, they need to correct that in the abstract. I suggest the authors mention in the abstract the outcomes of both patients, not just the microorganism characteristics.

Authors’ reply: We thank the reviewer for both comments. This is clarified in the abstract (lines 37 and 44 respectively)

Materials and Methods: I suggest the authors to explain more about the clinical outcomes such as hospitalization days, CBC findings, etc… They could provide a table for that.

Authors’ reply: the authors would like to thank the reviewer for the comment. Table 1 was added in the result section with the clinical profile as well as the hospitalization days (line 206).  

 In addition, do these patients have any basic immunological studies such as immunoglobulins and complement?

Authors’ reply: both patients were immunocompetent. This is added to the text (lines 200-201)

Discussion: There have been many publications about atypical presentations of meningococcal disease, such as conjunctivitis, sinusitis, among many others, I think the authors should point out those publications in their discussion.

Authors’ reply: We thank the reviewer for the comment. More information is provided in the discussion section lines 244-248.

Reviewer 2 Report

Comments and Suggestions for Authors

In this manuscript the authors have described two isolates identified as MenX ( 18, 25-44, F5-5, ST-823) and MenB (7-1, 1, F3-3, ST-7460; 32cc for PorA, FetA), respectively. Both isolates were susceptible to cefotaxime, ceftriaxone and rifampicin, while one of them expressed reduced susceptibility to penicillin and resistance to ciprofloxacin.

This work has been well conducted and described. 

Because Greece is a country with a relatively large transit of people (e.g., tourists) could the author expand the discussion on the possible origin of infection? 

Author Response

In this manuscript the authors have described two isolates identified as MenX ( 18, 25-44, F5-5, ST-823) and MenB (7-1, 1, F3-3, ST-7460; 32cc for PorA, FetA), respectively. Both isolates were susceptible to cefotaxime, ceftriaxone and rifampicin, while one of them expressed reduced susceptibility to penicillin and resistance to ciprofloxacin.

This work has been well conducted and described.

Because Greece is a country with a relatively large transit of people (e.g., tourists) could the author expand the discussion on the possible origin of infection?

Authors’ reply: We thank the reviewer for the comment. However, as both patients were of Greek origin, there was not a possibility to trace the origin of the infection.